# Prevalence and factors associated with tuberculosis among the mining communities in Mererani, Tanzania

Alexander W. Mbuya[1,2]◉*, Innocent B. Mboya[1,3]◉, Hadija H. Semvua[4]◉, Simon H. Mamuya[5]◉, Sia E. Msuya[1,3,6]◉

1 Department of Community Health, Institute of Public Health, Kilimanjaro Christian Medical University College (KCMUCo), Moshi, Tanzania, 2 Department of Community Health, Kibong'oto Infectious Diseases Hospital, Sanya Juu, Kilimanjaro, Tanzania, 3 Department of Epidemiology and Biostatistics, Institute of Public Health, Kilimanjaro Christian Medical University College (KCMUCo), Moshi, Tanzania, 4 Kilimanjaro Clinical Research Institute, Kilimanjaro Christian Medical Center, Moshi, Tanzania, 5 Department of Environmental and Occupational Health, School of Public Health and Social Sciences, Muhimbili University of Health and Allied Sciences, Dar es Salaam, Tanzania, 6 Department of Community Health, Kilimanjaro Christian Medical Center (KCMC), Moshi, Tanzania

◉ These authors contributed equally to this work.
* kiletsa13@gmail.com.

**Data Availability Statement:** All relevant data are within the manuscript and Supporting information files.

## Abstract

Tuberculosis (TB) is among diseases of global health importance with Sub Saharan Africa (SSA) accounting for 25% of the global TB burden. TB prevalence among miners in SSA is estimated at 3,000–7,000/100,000, which is about 3 to 10-times higher than in the general population. The study's objective was to determine the prevalence of TB and associated risk factors among mining communities in Mererani, northern Tanzania. This was a cross-sectional study conducted from April 2019 to November 2021 involving current Small Scale Miners (SSM) and the General Community (GC). A total of 660 participants, 330 SSM and 330 GC were evaluated for the presence of TB. Data were analysed using Statistical Package for the Social Sciences (SPSS) database (IBM SPSS Statistics Version 27.0.0.0). Binary logistic regression (Generalized Linear Mixed Model) was used to determine the association between TB and independent predictors. Prevalence of TB was 7%, about 24-times higher than the national prevalence of 0.295%. Participants from the general community had higher prevalence of TB 7.9% than SSM (6.1%). Both for SSM and the GC, TB was found to be associated with: lower education level (aOR = 3.62, 95%CI = 1.16–11.28), previous lung disease (aOR = 4.30, 95%CI = 1.48–12.53) and having symptoms of TB (aOR = 3.24, 95% CI = 1.38–7.64). Specifically for the SSM, TB was found to be associated with Human Immunodeficiency Virus (HIV) infection (aOR = 8.28, 95%CI = 1.21–56.66).

Though significant progress has been attained in the control of the TB epidemic in Tanzania, still hot spots with significantly high burden of TB exists, including miners. More importantly, populations surrounding the mining areas, are equally affected, and needs more engagement in the control of TB so as to realize the Global End TB targets of 2035.

**Funding:** AWM 91672520 DAAD German Academic Exchange Service https://www.daad.de/en/ The funders had no role in study design, data collection and analysis, decision to publish, or preparation of the manuscript.

**Competing interests:** The authors have declared that no competing interests exist.

## Introduction

Tuberculosis continues to be among the key global health challenges with SSA, which is also highly affected by the HIV epidemic, carrying the heaviest disease prevalence. In 2019, Africa accounted for 25% of the global TB prevalence, though the region has made a positive reduction in TB incidence and deaths by 16% and 19% respectively between 2015 to 2019. [1] While an estimated 25% of the global population is infected with Mycobacterium tuberculosis (*M.tb)*, several risk factors determines the chances of one to develop active TB, including HIV infection with 15% as a population attributable fraction of HIV infection as a driver of TB disease. [2] In the year 2020, Tanzania had TB notification rate of 142/100,000. [3]

The prevalence of TB among miners in Sub Saharan Africa is estimated to be between 3,000/100,000–7,000/100,000. [4–7] A recent report among gold miners in South Africa showed TB prevalence to be three-times higher compared to that in the general population, [8] while some countries have reported a TB prevalence among miners of up to 8 to 10-times higher relative to the general community. [4, 5, 9] Small-scale miners have relatively high risk of getting TB as a result of inadequate ventilation, significant exposure to respirable crystalline silica, HIV infection, in some cases poor nutrition and alcohol abuse. [6, 10] Tuberculosis transmission is favoured in poorly ventilated, dark and humid environment as is the case in most of the small-scale mining pits. As miners carry a huge TB prevalence, they also pose high risk to the surrounding communities they come into contact with. Noting the high prevalence of TB in the mining sector and its associated negative effects to the surrounding communities, the Heads of States in the Southern African Development Community (SADC) region came up with the '*Declaration of TB in the Mining Sector in Southern Africa*' in 2012. Among other issues, the declaration outlines the high contribution of the mining sector and the miners to the economy and development of the nations in the region but which comes at a huge negative health effects to the individuals miners and their families. [5]

Tanzania is among the nations visioning for zero new TB infections, zero stigma and zero deaths from TB, HIV and silicosis. The country is also committed to establishment of regular screening of miners and their close contact, referral of those presumed to have TB for laboratory diagnostic services and early initiation of treatment to those found to have TB. Locally at Mererani mines, the prevalence of TB among tanzanite miners has been reported to be well higher compared to the general population. [10] The SSM are generally all men, with no standard time of the daily working shifts, almost all don't use standard personal protective equipments and just a few will use locally improvised nose and mouth protective cloths while working. [11]

This study aimed to determine the current prevalence of TB among the mining communities in Mererani, situated in Simanjiro District, northern Tanzania and assess factors associated with TB. As per this study, the mining communities include two groups of population, the SSM (the mine workers who goes down the mining pits and perform the drilling activities) and the GC who are the peri-mining communities residing near the mines but engage in other non-mining activities. The study was necessary so as to provide the current burden of TB among the mining communities and elaborate on factors associated with occurrence of such burden.

## Materials and methods

### Study design and study site

This was a cross sectional study involving mining community. The main study titled 'Silica and radon exposures and its associated effects on respiratory system among small scale

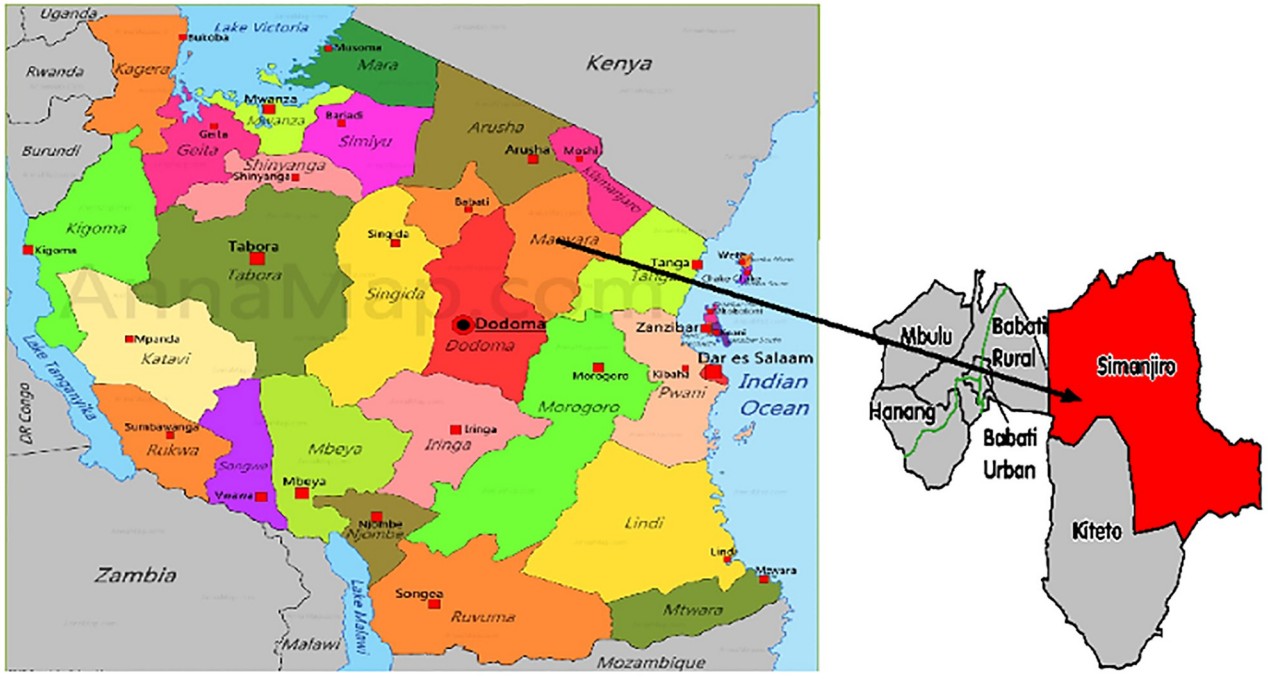

**Fig 1.**

tanzanite miners in Mererani'. The mining pits are located in Mererani (sometimes referred to as Mirerani) ward in Simanjiro District located in northern part of Tanzania (Fig 1). Usually the mine workers uses traditional knowledge and experience to track areas likely to have the gemstone, there is no professional exploration being done prior to deciding areas to dig the pits. With this approach one cannot estimate when and how much gemstone will be obtained. The study compared two groups of populations, SSM working in pits of the Mererani mines and the general community engaging in other non-mining activities, with no history of working in the mines but are residents of Mererani town, which is located about 5km from the mining pits. Mererani town is located about 70km from Arusha City and 20km from Kilimanjaro International Airport. To the best of our knowledge, Mererani it is the only place in the world where the tanzanite gemstone is being mined on commercial scale. Weather in Mererani is dry with short periods of rain and windy most of the time, making even the communities surrounding the mining pits to be at risk of mining related respiratory diseases. Mererani town has three health facilities (Mererani Health Center, Minja Dispensary and Roman Catholic Dispensary), the first two of which offers TB diagnostic and treatment services.

## Study population

Based on the National Census (2012), the population of Simanjiro district was around 178,693, with a district growth rate of 2.4%, the current population is estimated to be 221,211, with about half of them residing in Mererani ward. The Mererani mines are estimated to harbour around 9,000 SSM most of whom are constantly exposed to risky working conditions including exposure to mining dust. Though the pits owners provides daily meals for the SSM, but the SSM have no routine salaries, they enter into an agreement with the pit owner to receive a certain percentage of the earned money once a gemstone is found and sold, hence most SSM are

faced with social-economic hardship in most of the days. Individuals directly engaged with mining activities i.e. going down the pits are all men (women engage themselves with other activities like shop keeping and food vending in areas surrounding the mining pits). Hence, for the purpose of assessing the actual mining-related exposure to respirable crystalline silica and its' effects on the respiratory system (as per the main study requirements), only men were selected as study participants. At the time of data collection, there were 100 active mining licence holders (mining pit owner) registered under Manyara Region Miners Association, each with between 70 to 90 SSM, hence a maximum of 9,000 active SSM. The study targeted currently working SSM (those who goes down the pits and perform drilling work) and the GC engaging in other non-mining activities but residing near the mines who may be regarded as peri-mining communities. The general communities in Mererani are located about 5 kilometres from the mining pits.

## Sample size, sampling and data collection

The sample size of 660 study participants, 330 mine workers and 330 from the general community, was used. This was determined based on the estimated prevalence of silicosis among mine workers in Mererani mines of 16% as per clinical data from the Occupational Health Clinic Center (OHSC) at Kibong'oto Infectious Diseases Hospital (KIDH) and a number of outreach disease (including silicosis) screening programs conducted in Mererani mines by KIDH. Both at the OHSC and outreach services, all attended miners are investigated of silicosis regardless of presence or absence of symptoms. The minimum sample size was calculated using the formula,

$n = (z/d)^{2*}p(1 - p)$
where by:

$n$ = minimum sample size

z = critical value for 95% confidence interval (1.96)

d = degree of precision (0.05)

p = estimated prevalence of silicosis (16%)

This led to $n = 206.4$, which was then adjusted by +10% for lung function assessment, +10% for TB investigation and addition +30% for data loss, hence total sample came to 325, which was approximated to 330 SSM. The same number was taken for the general community to make the overall total sample of 660.

From the 100 active mining pits, 22 pits were randomly selected and one day per pit was reserved to elaborated about the study objectives to the SSM and the pits' managers. From each pit, 15 SSM were randomly selected guided by a coded list of miners, this was regarded as one cluster, making a total sample of 330 miners in 22 clusters. The number of 22 mining pits was selected so as to meet the required coverage on occupational exposure assessment (another nested study) and also to meet the required sample size of 330 mine workers who were to leave the workplace for disease evaluation at Kibong'oto Infectious Diseases Hospital for disease evaluation. This ensured enough number of workers are retained to avoid discontinuation of routine mining activities while at the same time meeting the required sample size of mine workers. Following discussion with pits' managers, it was agreed that 15 SSM was the highest number that could have left the working place at a given time without interfering the routine mining activities. These SSM were consented for lung health evaluation, testing for TB, silicosis, Impaired Lung Function, diabetes mellitus and HIV infection after which they were taken through the Interview Schedule which was adapted from Medical Research Council (MRC)

United Kingdom (UK) Respiratory Questionnaire, 1986. The 15 SSM from each of the mining pit formed one cluster as they were assumed to share similarities on the environmental (workplace) risk factors for TB, hence a total of 22 clusters for SSM.

For the GC in Mererani town, from a list of 85 streets, 22 streets were randomly selected. In each of the selected streets, houses (being it a shop, health facility or school) were selected consecutively, and from each selected house, individual participants were selected consecutively (as per inclusion and exclusion criteria) until 15 participants per street (cluster) were reached, making a total of 330 participants. For purposes of comparison with the miners, only men were selected as study participants. Then the participants were taken through the Interview Schedule.

The 15 randomly selected participants from each of the 22 mining pits and 22 houses in GC were scheduled in groups of 20 to 25 persons and provided with return transport to Occupational Health Service Center at Kibong'oto Infectious Diseases Hospital located in Siha District in Kilimanjaro Region, about 60 kilometres from Mererani town which at the time of the study it was the only nearby health facility with the required capacity to perform the study's investigations. This was done on different days so as to allow the routine work in specific pits to continue. Participants selected were educated on the importance of producing an early morning sputum sample for TB investigation and also a fasting blood sample for blood glucose tests. Hence, transport services were scheduled for the participants to be at the clinic before 0800am. At the Occupational Health Service Center, participants underwent general evaluation on body temperature, weight, oxygen saturation and blood pressure. This was followed by scheduled investigations and tests i.e. sputum collection and examination by GeneXpert MTB/ RIF (GeneXpert Dx System, Version 4.8, Cepheid, USA), blood samples for HIV tests using the rapid tests (SD BIOLINE HIV-1/2 3.0 Standard Diagnostic Inc.) and fasting blood glucose tests using GlucoPlus™ Blood Glucose Test Strips (Blood Glucose Meter GlucoPlus Inc. Quebec, CANADA) in which 1.5ul of whole blood was used and with a testing time of 15seconds. With investigation for TB, unlike with routine practice in which only the presumptive TB clients (those with at least one of the cardinal symptoms of TB i.e. cough, fever, excessive night sweating, weight loss and bloody stained sputum) do produce sputum, with this study all participants were asked to produce and provide one early morning (before brushing the teeth) sputum samples for laboratory tests. This was important not only because mining communities are among the high risk group for TB but also some reports have suggested presence of sub-clinical TB cases. [12, 13] The five cardinal symptoms as per the TB screening algorithm at community level, adapted from the National Tuberculosis and Leprosy Programme Manual for the Management of Tuberculosis and Leprosy in Tanzania [14] were asked and filled once for each study participant as part of the interview schedule to assist in further data analysis if required.

One chest X-ray (DRGEM Corporation, Korea) was done to each participant for the evaluation for silicosis and spirometry for evaluation of ILF was done to each participant using Easy on-PC (NDD Medizintechnik AG, Zurich, Switzerland). The digital chest X-rays were read and interpreted by a radiologist with significant experience in the field of occupational health using RADMAX CDR Digital Imaging and Communications (DICOM) viewer. The spirometer was calibrated twice per day, during the morning before commencing the tests and after performing 20 tests, using a 3L Volume Calibration Syringe (CRC Medical, DMS Limited, UK). All participants with spirometry results indicating obstruction were subjected to bronchodilator reversibility test (post-bronchodilator spirometry) in which the participants was given four puffs of bronchodilator (salbutamol inhaler) and spirometry was repeated 15 minutes after being given the medication. An increase of at least 200mls in Forced Expiratory Volume during the first second ($FEV_1$) and/or Forced Vital Capacity (FVC) was regarded as a

significant response, hence was diagnosed as asthma (and not COPD). Participants who failed to perform the initial spirometry test or could not produce reliable results, were re-scheduled for another test after one week until reliable results were obtained.

All the investigation results were documented in the data collection sheet (Microsoft Office Excel sheet). Participants found to have one or more of TB, HIV, silicosis, ILF or DM, were counselled to enrol to care and treatment services as per the guidelines by the Ministry of Health, Community Development, Gender, Elderly and Children (MoHCDGEC).

### Data management and analysis plan

The collected data were cleaned in Excel Sheet (Microsoft Excel for Mac, Version 16.42). Then the data were transferred to SPSS database (IBM SPSS Statistics Version 27.0.0.0) for analysis. Socio-demographic and clinical characteristics were presented in frequencies and percentage of occurrence in each of the two community groups i.e. SSM and the GC. The association between TB and presumed risk factors were determined by binary logistic regression (Generalized Linear Mixed Model) analyses. All methods were performed in accordance with the relevant guidelines and regulations. A number of predictor variables were transformed to suit certain criteria, these include age which was grouped into those of 18 to 40 years and those who are above 40 years as observations from clinical practice it has been noted most of miners found to have TB are 40 years and below. Marital status was grouped into those under any form of relationship and those without any form of relationship as this will assist in future plans on improving treatment adherence based on family-social support the TB patient is getting. Education level was grouped into those with above primary education level and those with primary education level and below. The duration of residing and work in the study areas was grouped into more than 5 years and those who have stayed for 5 years and below to account for accelerated/chronic silicosis and acute silicosis respectively. [15, 16] In addition, the overall median income for all 660 study participants was USD129.3 per month, hence the income levels per month were grouped into those with above the median value and those with the median value and below.

### Ethical approval and consent to participate

Ethical clearance to conduct the study was sought from the College Research and Ethical Review Committee of KCMUCo, No. 2416 and the Medical Research Coordinating Committee of the National Institute for Medical Research (NIMR), No: NIMR/HQ/R.8a/Vol.IX/3308. Permission to conduct the study was sought from the Permanent Secretary—Presidents' Office, Regional Administration & Local Government (PS-PORALG), the Regional Administrative Secretary (Manyara region), the District Executive Director (Simanjiro District) and Local Government Authorities in Mererani town and Hospital Administration at KIDH. Also, permission was sought from the owners/managers for the specific mining pits. Written informed consents, translated in Kiswahili language was obtained from each and all individual participants. The participants' identification was documented using numbers. Participants found to have TB and other diseases were treated for free at KIDH. Participants were provided with free return transport to OHSC at KIDH.

### Results

Table 1 shows the socio-demographic characteristics of the study participants who have been divided into two main groups i.e., the SSM and the GC. The two groups i.e., SSM and the GC deferred in a number of characteristics including age distribution with the SSM having statistically significantly lower median age of 35.0 (IQR 30.0–44.0) years compared to GC with a

**Table 1. Socio-demographic characteristics of the mining communities in Mererani (N = 660).**

| Characteristics | Total (N = 660) | Small Scale Miners (n = 330) | General Community (n = 330) | p value |
|---|---|---|---|---|
| **Age (years)** | | | | |
| Median (IQR) | | 35.0 (30.0–44.0) | 39.0 (32.0–46.0) | 0.002 |
| 18–40 | 412 (62.4) | 221 (67.0) | 191 (57.9) | 0.016 |
| >40 | 248 (37.6) | 109 (33.0) | 139 (42.1) | |
| **Marital Status** | | | | |
| Not in relationship | 112 (17.0) | 63 (19.1) | 49 (14.8) | 0.147 |
| In a relationship | 548 (83.0) | 267 (80.9) | 281 (85.2) | |
| **Education level** | | | | |
| ≤Primary | 468 (70.9) | 272 (82.4) | 196 (59.4) | <0.001 |
| >Primary | 192 (29.1) | 58 (17.6) | 134 (40.6) | |
| **Duration of Working/Living in Mererani (years)** | | | | |
| Median (IQR) | | 5.0 (4.0–7.0) | 8.5 (6.0–12.0) | <0.001 |
| ≤5[a] | 238 (36.1) | 179 (54.2) | 59 (17.9) | <0.001 |
| >5 | 422 (63.9) | 151 (45.8) | 271 (82.1) | |
| **Income (USD/month)** | | | | |
| Median (IQR) | | 86.2 (64.7–215.6) | 215.5 (129.3–344.8) | <0.001 |
| >129.3[b] | 326 (49.4) | 101 (30.6) | 225 (68.2) | <0.001 |
| ≤129.3 | 334 (50.6) | 229 (69.4) | 105 (31.8) | |

[a]The estimated cut-off duration of exposure to Respirable crystalline silica (RCS) associated with development of acute silicosis,
[b]The overall median income of the study participants, 1USD = 2,320TZS during data collection.

median age of 39.0 (IQR 32.0–46.0). Majority, 468 (70.9%) had education level of primary and below, of whom 272 (58.1%) were SSM. Among the SSM, 272 (82.4%) had education level of primary and below while for the GC it was 196 (59.4%). A total of 422 (63.9%) had duration of work in Mererani (residing and work at the study area) of more than 5years. The overall median duration of work in Mererani for all the participants is 5 years which is also taken as a cut-off duration for differentiating between acute and chronic silicosis. For the SSM, the median duration of work in Mererani was 5.0 (IQR 4.0–7.0) years which is shorter compared to GC with a median of 8.5 (IQR 6.0–12.0). The overall median income for all the participants was USD 129.3.

Table 2 shows the clinical characteristics of the study population. Of all the 660 subjects, 93 (14.1%) were smokers. The median durations of smoking for the two communities were 10.0 (IQR 4.0–15.8) years for SSM and 11.0 (IQR 5.0–17.0) years for the GC. Participants with history of lung diseases were 351 (53.2%). While among SSM the history of previous lung disease accounted for 205 (62.1%), it was 146 (44.2%) for the GC. In total, 279 (42.2%) met the criteria of being presumptive TB (had one or more among the five cardinal signs/symptoms of TB). Comparing the two populations, there were 186 (56.4%) and 93 (28.2%) presumptive TB subjects for the SSM and the GC respectively. Overall prevalence of TB was 7%. The GC had relatively higher prevalence of TB (7.9%) compared to the SSM (6.1%). Abnormal lung function (both obstructive and restrictive lung diseases) was found in 223 (33.8%). While among SSM the abnormal lung function accounted for 135 (40.9%), it was found among 88 (26.7%) of the GC. A total of 99 (15.0%) were found to have silicosis and all were SSM, of whom the prevalence of silicosis was 30.0%. Diabetes Mellitus was found in 37 (5.6%) of the study subjects. Those with DM among the SSM were 26 (7.9%) while among the GC they were 11 (3.3%).

**Table 2. Clinical characteristics of the mining communities in Mererani (N = 660).**

| Clinical Characteristics | Total (N = 660) | Small Scale Miners (n = 330) | General Community (n = 330) | p value |
|---|---|---|---|---|
| **Smoking status** | | | | |
| Median duration (years) | | 10.0 (4.0–15.8) | 11.0 (5.0–17.0) | 0.394 |
| Smokers | 93 (14.1) | 60 (18.2) | 33 (10.0) | 0.003 |
| Non-smokers | 567 (85.9) | 270 (81.8) | 297 (90.0) | |
| **Previous Lung Disease[a]** | | | | |
| Yes | 351 (53.2) | 205 (62.1) | 146 (44.2) | <0.001 |
| No | 309 (46.8) | 125 (37.9) | 184 (55.8) | |
| **Presumptive TB[b]** | | | | |
| Yes | 279 (42.3) | 186 (56.4) | 93 (28.2) | <0.001 |
| No | 381 (57.7) | 144 (43.6) | 237 (71.8) | |
| **Tuberculosis** | | | | |
| Yes | 46 (7.0) | 20 (6.1) | 26 (7.9) | 0.360 |
| No | 614 (93.0) | 310 (93.9) | 304 (92.1) | |
| **Lung Function** | | | | |
| Abnormal | 223 (33.8) | 135 (40.9) | 88 (26.7) | <0.001 |
| Normal | 437 (66.2) | 195 (59.1) | 242 (73.3) | |
| **HIV status** | | | | |
| Yes | 8 (1.2) | 6 (1.8) | 2 (0.6) | 0.176 |
| No | 652 (98.8) | 324 (98.2) | 328 (99.4) | |
| **Silicosis status** | | | | |
| Yes | 99 (15.0) | 99 (30.0) | 0 (0.0) | NA |
| No | 561 (85.0) | 231 (70.0) | 330 (100.0) | |
| **Diabetes Mellitus** | | | | |
| Yes | 37 (5.6) | 26 (7.9) | 11 (3.3) | 0.014 |
| No | 623 (94.4) | 304 (92.1) | 319 (96.7) | |

[a]Include tuberculosis, bacterial pneumonia and/or silicosis,

[b]Participant with one or more among: productive cough for at least two weeks, fever for at least two weeks, bloody stained sputum, weight loss of at least 3kg in one month or excessive night sweating.

NA means not applicable.

Table 3 shows the Crude Odds Ratios (cOR) and Adjusted Odd Ratios (aOR) from Binary Logistic Regression under the Generalized Linear Mixed Models (GLMM). From the aOR, three predictors were found to have significant association with TB i.e., education, history of previous lung disease and being presumptive TB. The SSM had about 55% lower odds of having TB as compared to the GC (aOR = 0.45, 95%CI = 0.15–1.32) though the findings was not statistically significant. Both education level and previous lung disease were found to have statistically significant associations with TB, with those having lower education level (aOR = 3.82, 95%CI = 1.32–11.08) and those with previous lung disease (aOR = 3.86, 95%CI = 1.33–11.25) having almost 4-times higher odds of having TB compared to subjects with above primary education level and not having previous lung disease respectively. Similarly, presumptive TB subjects had more than 3-times higher odds of having TB compared to the none presumptive TB participants (aOR = 3.26, 95%CI = 1.38–7.65).

Table 4 shows the analysis of factors associated with TB stratified by subjects' occupation i.e. SSM and GC. Upon the stratification, of all the predictors, HIV was found to have a statistically significant association with TB among the SSM with about more than 8-times higher

**Table 3. Factors associated with tuberculosis among mining communities in Mererani (N = 660).**

| Variable | cOR (95%CI)[*] | p value | aOR (95%CI)[*] | p value |
|---|---|---|---|---|
| **Population** | | | | |
| Mine workers | 0.75 (0.41–1.38) | 0.360 | 0.45 (0.15–1.32) | 0.146 |
| Gen. Community | | | | |
| **Age (years)** | | | | |
| 18–40 | 0.58 (0.32–1.06) | 0.074 | 0.76 (0.34–1.69) | 0.503 |
| >40 | | | | |
| **Marital Status** | | | | |
| Not in relationship | 0.87 (0.38–2.00) | 0.743 | 1.85 (0.67–5.12) | 0.233 |
| In relationship | | | | |
| **Education level** | | | | |
| ≤Primary | 3.59 (1.40–9.23) | 0.008 | 3.82 (1.32–11.08) | 0.014 |
| >Primary | | | | |
| **Duration of Living in Mererani (years)** | | | | |
| ≤5[a] | 0.47 (0.23–0.97) | 0.040 | 0.94 (0.35–2.52) | 0.904 |
| >5 | | | | |
| **Income (USD/month)** | | | | |
| >129.3[b] | 1.03 (0.56–1.87) | 0.932 | 0.99 (0.44–2.24) | 0.983 |
| ≤129.3 | | | | |
| **Smoking** | | | | |
| Smokers | 1.54 (0.72–3.30) | 0.272 | 2.32 (0.93–5.83) | 0.072 |
| Non Smokers | | | | |
| **Previous Lung Disease** | | | | |
| Lung disease | 6.50 (2.71–15.54) | 0.001 | 3.86 (1.33–11.25) | 0.013 |
| No lung disease | | | | |
| **Presumptive TB** | | | | |
| Presumptive | 4.83 (2.40–9.68) | 0.001 | 3.26 (1.38–7.65) | 0.007 |
| Not presumptive | | | | |
| **Lung Function (LF)** | | | | |
| Impaired LF | 1.41 (0.77–2.60) | 0.266 | 1.08 (0.52–2.22) | 0.840 |
| Normal LF | | | | |
| **HIV Infection** | | | | |
| HIV infected | 4.61 (0.90–23.49) | 0.066 | 5.77 (0.74–44.78) | 0.093 |
| Not HIV infected | | | | |
| **Silicosis** | | | | |
| Silicosis | 0.68 (0.26–1.75) | 0.419 | 0.79 (0.25–2.50) | 0.693 |
| No silicosis | | | | |
| **Diabetes Mellitus (DM)** | | | | |
| DM | 0.75 (0.18–3.23) | 0.701 | 0.51 (0.10–2.58) | 0.418 |
| No DM | | | | |

[*] cOR and aOR based on Generalized Linear Mixed Models—Binary Logistic Regression Analysis,

[a] The estimated cut-off duration of exposure to Respirable Crystalline Silica (RCS) associated with development of acute silicosis,

[b] The overall median income of the study participants, 1USD = 2,320TZS during data collection.

**Table 4. Factors associated with tuberculosis among mining communities: Stratified by occupation (N = 660).**

| Variable | Small Scale Miners (330) | | General Community (330) | |
|---|---|---|---|---|
| | aOR (95%CI)[*] | p value | aOR (95%CI)[*] | p value |
| **Age (years)** | | | | |
| 18–40 | 0.98 (0.34–2.81) | 0.970 | 0.44 (0.10–1.87) | 0.268 |
| >40 | | | | |
| **Marital Status** | | | | |
| Not in relationship | 1.72 (0.58–5.11) | 0.324 | 1.70 (0.11–26.23) | 0.704 |
| In relationship | | | | |
| **Education level** | | | | |
| ≤Primary | 1.43 (0.37–5.52) | 0.606 | 8.02 (1.39–46.23) | 0.020 |
| >Primary | | | | |
| **Duration of Working/Living in Mererani (years)** | | | | |
| ≤5[a] | 0.41 (0.13–1.28) | 0.124 | 3.00 (0.34–26.43) | 0.322 |
| >5 | | | | |
| **Income (USD/months)** | | | | |
| >129.3[b] | 0.82 (0.28–2.40) | 0.711 | 2.33 (0.50–10.91) | 0.280 |
| ≤129.3 | | | | |
| **Smoking** | | | | |
| Smokers | 1.68 (0.61–4.61) | 0.316 | 5.41 (0.48–60.15) | 0.169 |
| Non-Smokers | | | | |
| **Previous Lung Disease** | | | | |
| Lung disease | 1.55 (0.50–4.82) | 0.448 | 3.37 (0.38–29.97) | 0.275 |
| No lung disease | | | | |
| **Presumptive TB** | | | | |
| Presumptive | 0.60 (0.22–1.60) | 0.304 | 58.84 (5.40–641.32) | 0.001 |
| Not presumptive | | | | |
| **Lung Function** | | | | |
| Impaired LF | 1.16 (0.46–2.93 | 0.746 | 1.37 (0.32–5.83) | 0.672 |
| Normal LF | | | | |
| **HIV Infection** | | | | |
| HIV infected | 8.31 (1.22–56.35) | 0.030 | | |
| Not HIV infected | | | | |
| **Silicosis** | | | | |
| Silicosis | 0.86 (0.33–2.27) | 0.757 | | |
| No silicosis | | | | |
| **Diabetes Mellitus** | | | | |
| DM | 0.60 (0.10–3.57) | 0.576 | 0.25 (0.02–3.95) | 0.324 |
| No DM | | | | |

[*]aOR based on Generalized Linear Mixed Models—Binary Logistic Regression Analysis,

[a]The estimated cut-off duration of exposure to Respirable Crystalline Silica (RCS) associated with development of acute silicosis,

[b]The overall median income of the study participants.

odds of TB among the HIV infected as compared to non-HIV infected subjects (aOR = 8.31, 95%CI = 1.22–56.35). For the GC group, a number of predictors were found to have statistically significant association with TB, including education level with more than 8-times higher odds of TB among those with primary and below education (aOR = 8.02, 95%CI = 1.39–46.23). Presumptive TB subjects had almost 60-times higher odds of having TB relative to the

non-presumptive subjects, but the precision was low (wide confidence interval). Two subjects were found to have HIV infection, all of whom did not have TB. None of the subjects from the GC was found to have silicosis.

## Discussion

Overall prevalence of TB was 7,000/100,000 (about 7.0%) which is about 24-times higher the national (Tanzania) prevalence of 295/100,000. [17] This could to a large extent be attributed to the presumed poor ventilation within the Mererani mines (a factor which was not assessed in this study) coupled with lack of systematic screening of TB to all mine workers, which in turn facilitate continuous TB transmission within the mining pits. It is widely reported that more men suffers TB compared to women, including reports from Tanzania which shows about 60% of all TB patients are men. [18] Since this study involved only men participants, this could also explain the significantly high prevalence of TB in this community. Prevalence of TB for both sub-populations i.e. SSM and GC are well above those reported among miners from other studies of around 6-times higher than in the general population. [6] Other studies reported slightly different TB prevalence in mine workers: 6.6% among SSM in Mererani [19] and 9.5% among copper miners in Zambia. [20] In general, the current study findings shows the burden of TB among SSM and the communities surrounding the SSM to be very high. A cross-sectional study among migrant mine worker in Mozambique reported smaller TB prevalence of around 0.3%. [21] But many of the mine workers in Mozambique migrate and work in South Africa where there is a well-established health system that include systematic screening of TB among the mine workers which could explain the low TB prevalence in this community.

Though both SSM and the GC had significantly higher burden of TB relative to the national prevalence, but unexpectedly the GC had higher TB prevalence of 7.9% (which is about 31-times higher the national prevalence) compared to the SSM with 6.1% (which is about 24-times higher the national prevalence). This findings is of significant public health importance as traditionally it has been reported miners to have relatively higher prevalence of TB compared to the GC. [6, 7] This raises concern on possibility of other, yet unknown risk factor(s) among the GC that may predispose them to the high burden of TB. While the risk of TB transmission from the SSM to the GC exist, some study in South Africa have reported TB transmission to occurs mostly within specific community groups (like mine workers themselves) rather than across the different community groups. [22] These findings could not apply for SSM in Mererani given the different levels of mechanization and industrialization between the two countries, which in turn determine the effectiveness of the health systems in the workplace. For the past 5 years (after noting the high TB burden among the SSM in Mererani from hospital clinical data), targeted interventions have been conducted by Kibong'oto Infectious Diseases Hospital (KIDH), a specialized referral hospital for infectious diseases which has been providing health services for the miners for more than 40 years. These interventions to SSM involves provision of health education, sensitization, screening and investigation of TB among the mine workers in Mererani (that was not being done in the GC) and have been conducted using the 'contact tracing and investigation approach' in which when a miner visiting KIDH is found to have TB, the mining pit and miners from which the TB patient has been found is targeted for these health services. Through this approach, miners in 65 of the 100 active mining pits have been provided with health services. While to the best of our understanding there is no such programs targeting the community surrounding the mines, these communities are also faced with other key risk factors including poor housing (huts with small windows which are mostly permanently covered with boxes and/or iron sheets, hence low ventilation and light

within the huts). As the miners interact with the GC in various social events on daily basis, it is likely that cross-transmission of TB between the two population occurs. Given the poor housing condition and lack of targeted health services in the GC, any person with active TB will have a higher chance of transmitting the disease to the rest of the family members. This could also explain the observed high prevalence of TB among GC relative to the SSM. While TB remains to be among the top global health challenge, both on morbidity and mortality, having strategies to addressing TB among these communities is of paramount importance if we are to realize the End TB targets.

After working shift, the SSM goes out of the working premises to their temporary residencies in Mererani town and occasionally will go to visit their families away from Mererani town. The overall high burden of TB among the mining communities poses additional risk of TB transmission to other household members but might also perpetuates recurrency of TB disease which in turn has been associated with poor treatment outcomes. [10, 22] Hence more efforts and resources on prevention and control of TB need to be fairly channelled towards the GC in Mererani, but also determining the prevalence of TB among family members and GC in the SSM home residencies.

While the SSM formed 66.7% of all presumptive TB subjects, they accounted for only 43.5% of all TB patients. The presumptive TB criteria was found to be positively related to the burden of TB for the GC but inversely related to the burden of TB for the SSM. Despite the higher burden of HIV and Diabetes Mellitus among the SSM, which have traditionally known to be risk factors for TB, but the SSM had higher levels of previous lung disease (including TB and chronic obstructive pulmonary disease), which to a large extent have symptom similar to those of pulmonary TB. This could explain the finding of majority (66.7%) of presumptive TB being SSM but only to account for 43.5% of TB patients. This suggests the need of a different and more realistic set of signs/symptom to be used as a TB screening criterion for the SSM. In addition, the presumptive TB under the GC community had relatively small number of parameters required to meet the criteria of presumptive TB i.e., the TB signs/symptoms. This finding challenges the current clinical practice on screening of TB based on the symptomatic approach as this could be associated with missing a significant number of TB patients, setting back the End TB milestones. A study by Mtei et al. [23] conducted in Dar es Salaam, Tanzania reported 29% of diagnosed TB patients among HIV infected individuals had sub-clinical presentation.

For several decades, Mererani mines have provided livelihood for many families, especially in the northern part of Tanzania. It has been one of the utmost choices for many young people who could not proceed with the formal education after completion of primary education. Hence, among the SSM relatively large proportion of young-aged individuals engage in mining activities compared to the older ones. This has been reflected from the finding that statistically significant proportion of the subject found to have TB were among those with primary education level and below. The finding could be associated with low awareness and understanding of TB and the importance of acting positively to control the disease. While other TB control mechanisms continue to be implemented, adolescent focused health education and behaviour changing program have to be designed and effected as additional measures.

Among those found to have TB in the current study, 87.0% had history of previous lung disease, of which included TB. It has been found that previous patients of TB to have increased risk of recurrent episodes of TB. [24] Another study reported a TB prevalence of 7.5% [21] though this report based only on those with previous history of TB.

While for the GC being presumptive TB was positively associated with having TB, such association was negative for the SSM with about 70% less chance of the presumptive TB participants to have TB. For at least the past four years, a significant proportion of the SSM in Mererani have been subjected to a number of TB screening and investigation programmes (after

being noted to have high burden of TB) but also unpublished clinical data at KIDH have shown around 40 to 45% of SSM from Mererani to have Post-Tuberculosis Pulmonary Conditions (PTPC) including Chronic Obstructive Pulmonary Diseases (COPD). In addition, from the current study it has been found that almost 43% of the SSM were found to have silicosis. Usually, both of these conditions i.e. PTPC and silicosis have signs/symptoms related to TB, hence could confound the association between being presumptive TB and having TB. This calls for determination of a more effective set of criteria for TB screening among the SSM.

The current study found the prevalence of TB among those with HIV infection to be 25%, equivalent to 4-times higher odds of TB among the HIV infected participants but the association was not statistically significant. But upon stratification by occupation, the HIV-infected SSM were found to have more than 8-times higher odds of having TB relative to the HIV-negative subjects. Other studies have reported varying TB prevalence among HIV infected individuals. A cross sectional hospital-based study in Arba Minch, Ethiopia reported a prevalence of 7.2% [25] while another retrospective review analysis conducted at Hawassa University Referral Hospital in Ethiopia reported a prevalence of 18.2%. [26] The current finding does tally with well-established findings of an association between TB and HIV, which not only increases the risk of TB, but also it has been associated with poor treatment outcomes among TB patients. [4, 10]

Of note, the prevalence of HIV infection among SSM (1.8%) was three-times higher compared to that among GC (0.6%), both of which and the overall HIV infection prevalence (SSM & GC combined) of 1.2% are below the national average of 5.0%. [27] This could be attributed to the on-going TB control programmes among the mining communities which include HIV control strategies.

Of all the participants, 93 (14.1%) were smokers, of whom SSM accounted for 60 (64.5%). Unlike other reports showing an association between TB and smoking, [20] this study showed the smokers to have about 1.8-times higher odds of having TB but the association not statistically significant. Some studies' reports have associated TB with silicosis, [4, 9, 20] but the overall analysis from this study could not find any such association. With about a quarter of the global population estimated to have latent TB infection, both HIV and silicosis have been reported to be important risk factors for re-activation of *M.tb* infection into active diseases among mine workers. [22]

The study successfully met its aim, and the findings will significantly contribute on providing up to date status of the TB situation but also provide important recommendations to the government and other stakeholders on the efforts to end the TB epidemic in the country, as part of the Global efforts to End TB by 2035.

Study limitations included: The study could not describe the patterns of mix-up between the SSM and the GC, of which could be impacting the transmission of TB between the two communities. Some residents of Mererani, especially the SSM have a tendency of leaving the area to seek family support once they fall sick. The study could not be extended to capture information and assess presence of disease among those subjects; hence this could have led to some variations in the observed findings. Though not directly involved in mining activities (going down the pits), women are found in the mining areas engaging in other activities but may also be affected. But due to the inclusion criteria, this study failed in include women.

## Conclusions

This study found the prevalence of TB among the mining communities to be very high (relative to the national prevalence), but more importantly and unexpectedly, the study found the prevalence of TB among the general community (peri-mining community) to have even

higher prevalence of TB, more than the miners. Mining is among the fast growing sectors in Tanzania, involving several types of minerals/gemstones with a current estimation of more than 1.5 million SSM [19, 28] hence the chances of TB transmission and its impact in the community could be even greater. This calls for going beyond the traditional approach of targeting the miners more than the surrounding communities in the control of TB. If we have to realize the Global End TB targets, more efforts on TB control have to be put in this and similar communities with relatively hight disease burden. More studies have to be done to assess the determinants of such high TB burden in the mining communities, including dynamics of TB transmission.

## Supporting information

**S1 File.**
(ZIP)

## Acknowledgments

We would like to acknowledge all professors, tutors and lecturers from the KCMUCo Institute of Public Health for their precious time on imparting new knowledge on us and making close follow up on our academic progress. Our sincere thanks to the staff of OHSC at Kibong'oto Hospital for their support and much of the clinical work. Last but not least our appreciations goes to the members of the Technical Working Group of the TB in the mining sector in Tanzania for such good advice and support in this work.

## Author Contributions

**Conceptualization:** Alexander W. Mbuya, Hadija H. Semvua.

**Data curation:** Alexander W. Mbuya, Sia E. Msuya.

**Formal analysis:** Alexander W. Mbuya, Innocent B. Mboya, Simon H. Mamuya, Sia E. Msuya.

**Funding acquisition:** Alexander W. Mbuya.

**Investigation:** Alexander W. Mbuya.

**Methodology:** Alexander W. Mbuya, Innocent B. Mboya, Hadija H. Semvua, Simon H. Mamuya, Sia E. Msuya.

**Project administration:** Alexander W. Mbuya, Hadija H. Semvua, Sia E. Msuya.

**Resources:** Alexander W. Mbuya, Sia E. Msuya.

**Software:** Alexander W. Mbuya, Sia E. Msuya.

**Supervision:** Alexander W. Mbuya, Innocent B. Mboya, Hadija H. Semvua, Simon H. Mamuya, Sia E. Msuya.

**Validation:** Alexander W. Mbuya, Simon H. Mamuya, Sia E. Msuya.

**Visualization:** Alexander W. Mbuya, Hadija H. Semvua, Simon H. Mamuya, Sia E. Msuya.

**Writing – original draft:** Alexander W. Mbuya.

**Writing – review & editing:** Alexander W. Mbuya, Innocent B. Mboya, Simon H. Mamuya, Sia E. Msuya.

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
