## [Decision Letter · Decision Letter 0]

9 Mar 2022

PONE-D-22-00417Prevalence and factors associated with tuberculosis among the mining communities in Mererani, TanzaniaPLOS ONE

Dear Dr. Mbuya,

Thank you for submitting your manuscript to PLOS ONE. After careful consideration, we feel that it has merit but does not fully meet PLOS ONE’s publication criteria as it currently stands. Therefore, we invite you to submit a revised version of the manuscript that addresses the points raised during the review process.

We look forward to receiving your revised manuscript.

Kind regards,

Qigui Yu, M.D./Ph.D

Academic Editor

PLOS ONE

Journal Requirements:

3. Please upload a copy of Supporting Information S1 Text. Dataset, S2 Text. Cover letter, S3 Text. Study protocol, S4 Text. Abstract, which you refer to in your text on page 26.

Reviewers' comments:

Reviewer's Responses to Questions

**Comments to the Author**

1. Is the manuscript technically sound, and do the data support the conclusions?

Reviewer #1: Partly

Reviewer #2: Yes

2. Has the statistical analysis been performed appropriately and rigorously? 

Reviewer #1: No

Reviewer #2: Yes

3. Have the authors made all data underlying the findings in their manuscript fully available?

Reviewer #1: No

Reviewer #2: Yes

4. Is the manuscript presented in an intelligible fashion and written in standard English?

Reviewer #1: Yes

Reviewer #2: Yes

5. Review Comments to the Author

Reviewer #1: The reported study was designed to estimate the prevalence of tuberculosis among miners and non-miners in a specific geographical area in northern Tanzania. The design selected for this purpose appears to be a case -control in which randomly selected miners from 22 randomly selected mining pits were compared with non-miner who are residents of a nearby town. The study was conducted over a period of about one and a half years ( April 2019 to November 2020). A total of 660 people were recruited into the study. The overall prevalence of TB was 7% with non-miners having a higher prevalence of TB at 7.9% compared with miners who had a TB prevalence of 6.1%. Risk factor analysis revealed a lower education level, previous lung disease, symptoms compatible with TB and HIV infection to be associated with the presence of active TB. The finding of a higher prevalence of TB in non- miners compared with miners was a surprising finding in this study.

There are major issues with this study which include

1. Lack of details on how the sample size was determined. Why was it decided to select 22 of 100 mining pits?

2. Lack of information on the TB screening and testing algorithm

3. Lack of clarity on the number of times the study participants were screened and or tested for TB. Were study participants repeat screened and or tested over the period of the study? Additionally, the algorithm for TB screening and or testing is not entirely clear.

4. Lack of information on how silicosis was diagnosed. It appears chest x-rays were not done or if they were done, this is not mentioned in the paper.

5. Lack of information on how lung function testing was carried out

6. In a cross – sectional study design why was matching not carried out?

There are also a few minor issues which include

1. Line 8 appears to have an error – the statement says that HIV infection accounts for 15% of HIV related TB. I suppose what the authors want to say is that the population attributable fraction of HIV as a driver of TB disease is 15%

2. Lines 60-62 appears irrelevant to the subject matter of the paper. There are many other statements in the paper (an example is the statement that says “the mining owner hires a manager) that can be deleted because they add very little to the value of the paper.

3. There are discrepancies in the figures provided for the size of the mining population. In line 123, the figure provided is 12,000 while in lines 134 -135 the figure provided is 9,000

4. In the acknowledgement section the corresponding author uses the first person singular (I) to acknowledge those who supported the study , making the reader wonder what happened to the partnership with the other authors in the design, implementation and reporting of this research work.

Reviewer #2: In this study, Mbuya et al. evaluated TB prevalence and several associated risk factors among 330 Small Scale Miners (SSM) and 330 General Community (GC) from the mining communities in Mererani, Tanzania. They found that both SSM and GC had a very high TB prevalence of 7% and 7.9%, respectively, which was approximately 27% higher than the national prevalence. Lower education level, previous lung diseases, and having TB symptoms were associated with TB in the combined participants. However, only HIV infection was associated with TB among the SSM. Overall, the study was well done, and the manuscript was well written. There are only several minor concerns/questions.

Why the mining communities in Mererani have a very high TB prevalence relative to the national average? Is there a gender difference in the national TB prevalence?

Line 58: It is not clear what “about 15% of HIV related TB disease” means.

Line 222: SSM with a median age of 35 is not “much” younger than GC with a median age of 39.

For Tables 1 and 2, please add p values for comparisons between SSM and GC. In Tables 1 and 4, should “Duration of Living” be replaced with Duration of Working/Living?

Lines 256-258 indicate 6 significant predictors. However, there are only 3 with a p value < 0.05 in Table 3.

Line 291: 6980/100,000 should be 6970/100,000.

6. PLOS authors have the option to publish the peer review history of their article (what does this mean?). If published, this will include your full peer review and any attached files.

Reviewer #1: **Yes: **Jeremiah Chakaya

Reviewer #2: No

---

## [Author Response · Author response to Decision Letter 0]

21 Apr 2022

Reviewer #1:

1. Lack of details on how the sample size was determined. Why was it decided to select 22 of 100 mining pits?

Response: Line 148 – 156 

The number of 22 mining pits was selected so as to meet the required coverage on occupational exposure assessment (another nested study) and also to meet the required sample size of 330 mine workers who were to leave the workplace for disease evaluation at Kibong’oto Infectious Diseases Hospital for disease evaluation. This ensured enough number of workers are retained to avoid discontinuation of routine mining activities while at the same time meeting the required sample size of mine workers. Following discussion with pits’ managers, it was agreed that 15 SSM was the highest number that could have left the working place at a given time without interfering the routine mining activities

2. Lack of information on the TB screening and testing algorithm

Response: Line 192 - 202

With investigation for TB, unlike with routine practice in which only the presumptive TB clients (those with at least one of the cardinal symptoms of TB i.e. cough, fever, excessive night sweating, weight loss and bloody stained sputum) do produce sputum, with this study all participants were asked to produce and provide one early morning (before brushing the teeth) sputum samples for laboratory tests. This was important not only because mining communities are among the high risk group for TB but also some reports have suggested presence of sub-clinical TB cases. The five cardinal symptoms as per the TB screening algorithm at community level (adapted from the National Tuberculosis and Leprosy Programme Manual for the Management of Tuberculosis and Leprosy in Tanzania, April 2020) were asked and filled once for each study participant as part of the interview schedule to assist in further data analysis if required.

3. Lack of clarity on the number of times the study participants were screened and or tested for TB. Were study participants repeat screened and or tested over the period of the study? Additionally, the algorithm for TB screening and or testing is not entirely clear.

Response: Line 196, 198 – 202

As this was a cross sectional study, each participant was screened (using the interview schedule) once and each was tested for TB once. The screening algorithm was adapted from National Tuberculosis and Leprosy Programme Manual for the Management of Tuberculosis in Tanzania, April 2020 which is based on assessing presence of any among the five cardinal symptoms of TB i.e. cough, fever, excessive night sweating, weight loss and bloody stained sputum. But this screening was done only for the purpose of later to assess the association between having TB and being a presumptive TB (a person with at least one of the five symptoms) client. All study participants produce sputum for TB investigation, regardless of being a presumptive TB or not.

4. Lack of information on how silicosis was diagnosed. It appears chest x-rays were not done or if they were done, this is not mentioned in the paper.

Response: Line 203 – 204 

Silicosis was diagnosed by a radiologist with significant experience in the area of occupational lung diseases, including mine workers. One chest X-ray was done to each participant using a digital X-ray machine.

5. Lack of information on how lung function testing was carried out

Response: Line 204 – 217

Lung function tests were done to each participant using Easy on-PC with twice per day calibration, during the morning before commencing the tests and after performing 20 tests, using a 3L Volume Calibration Syringe.

6. In a cross – sectional study design why was matching not carried out?

Response: Line 106 

This was a cross sectional study aiming to determine the prevalence of TB and associated risk factors being among all 660 participants. The stratification between the mine workers (SSM) and the peri-mining communities (GC) was just to assess on how the communities differ as they have varying occupations. Hence matching was not required.

There are also a few minor issues which include:

1. Line 8 appears to have an error – the statement says that HIV infection accounts for 15% of HIV related TB. I suppose what the authors want to say is that the population attributable fraction of HIV as a driver of TB disease is 15%

Response: Line 58 – 59 

Upon reviewing the manuscript, I found the referred statement was on line 58. I have rephrased the sentence to be clearer as per reviewer’s comment.

2. Lines 60-62 appears irrelevant to the subject matter of the paper. There are many other statements in the paper (an example is the statement that says “the mining owner hires a manager) that can be deleted because they add very little to the value of the paper.

Response: Line 60

The statements have been deleted as per reviewer’s comment.

3. There are discrepancies in the figures provided for the size of the mining population. In line 123, the figure provided is 12,000 while in lines 134 -135 the figure provided is 9,000

Response: Line 127 & 140

The figure has been corrected. Was a typing error and the correct figure is 9,000.

4. In the acknowledgement section the corresponding author uses the first person singular (I) to acknowledge those who supported the study , making the reader wonder what happened to the partnership with the other authors in the design, implementation and reporting of this research work.

Response: Line 475 – 480

The paragraph has been reviewed and appropriate correction made as per reviewer’s comment.

Reviewer #2:

General response: Line 289 – 291

The TB prevalence of 7% was the general prevalence from all 660 (SSM and GC) participants. The SSM had a TB prevalence of 6.1% while the GC had a TB prevalence of 7.9%.

Why the mining communities in Mererani have a very high TB prevalence relative to the national average? Is there a gender difference in the national TB prevalence?

Response: Line 342 – 349

This could to a large extent be attributed to the presumed poor ventilation within the Mererani mines (a factor which was not assessed in this study) coupled with lack of systematic screening of TB to all mine workers, which in turn facilitate continuous TB transmission within the mining pits. It is widely reported that more men suffers TB compared to women, including reports from Tanzania which shows about 60% of all TB patients are men. Since this study involved only men participants, this could also explain the significantly high prevalence of TB in this community.

Line 58: It is not clear what “about 15% of HIV related TB disease” means.

Response: Line 58 – 59

This means that the population attributable fraction of HIV as a driver of TB disease is 15%

Line 222: SSM with a median age of 35 is not “much” younger than GC with a median age of 39.

Response: Line 264 – 267 

The phrase has been corrected by replacing with ‘statistically significantly lower’.

For Tables 1 and 2, please add p values for comparisons between SSM and GC. In Tables 1 and 4, should “Duration of Living” be replaced with Duration of Working/Living?

Response: Table 1&2 

The p values have been added and the phrase “Duration of Living” replaced with “Duration of Working/Living”

Lines 256-258 indicate 6 significant predictors. However, there are only 3 with a p value < 0.05 in Table 3.

Response: Line 303 – 304

The phrase has been corrected to only the three predictors

Line 291: 6980/100,000 should be 6970/100,000.

Response: Line 341

As the overall prevalence was 7%, this equates to 7,000/100,000.

Note: Line number being referred here are as per the revised manuscript with track changes.

---

## [Decision Letter · Decision Letter 1]

10 Jun 2022

PONE-D-22-00417R1Prevalence and factors associated with tuberculosis among the mining communities in Mererani, TanzaniaPLOS ONE

Dear Dr. Alexander William Mbuya,

Thank you for submitting your manuscript to PLOS ONE. After careful consideration, we feel that it has merit but does not fully meet PLOS ONE’s publication criteria as it currently stands. Therefore, we invite you to submit a revised version of the manuscript that addresses the points raised during the review process.

We look forward to receiving your revised manuscript.

Kind regards,

Qigui Yu, M.D./Ph.D

Academic Editor

PLOS ONE

Reviewers' comments:

Reviewer's Responses to Questions

**Comments to the Author**

1. If the authors have adequately addressed your comments raised in a previous round of review and you feel that this manuscript is now acceptable for publication, you may indicate that here to bypass the “Comments to the Author” section, enter your conflict of interest statement in the “Confidential to Editor” section, and submit your "Accept" recommendation.

Reviewer #1: (No Response)

Reviewer #2: (No Response)

2. Is the manuscript technically sound, and do the data support the conclusions?

Reviewer #1: Partly

Reviewer #2: (No Response)

3. Has the statistical analysis been performed appropriately and rigorously? 

Reviewer #1: Yes

Reviewer #2: (No Response)

4. Have the authors made all data underlying the findings in their manuscript fully available?

Reviewer #1: Yes

Reviewer #2: (No Response)

5. Is the manuscript presented in an intelligible fashion and written in standard English?

Reviewer #1: No

Reviewer #2: (No Response)

6. Review Comments to the Author

Reviewer #1: While the authors have responded to all the issues raised in the previous reviews, there are still a number of issues that need to be addresses. These include:

1. Line 59- the authors may wish to use the most recent TB notification data from the 2021 WHO global TB report

2. Line 92-97- it is advised that the statement be reworded . The study aimed at determining the prevalence of TB and the risk factors for disease in the studied population. It is does not appear appropriate to include the statement that says the study met its aims and will feed into policy. This issue of whether the study met its aims etc can be shifted to the discussion and conclusion section.

3. The findings of the study are significantly different from those of multiple other studies on TB in miners. The issue is not sufficiently discussed. The hypothesis put forward is that there have been a TB screening program among the miners that may have reduced the burden of TB in this population compared to the surrounding community. It is not indicated how long the TB screening program has gone on and what the coverage has been. To begin to see a drop in the burden of TB in a population arising from TB screening and testing services, the screening needs to reach a high proportion of the target population and likely be repeated many times over time. While this issue was not raised in the previous review, it seems important enough to be raised now. The study found a rate of TB in those engaged directly in mining work that was lower than in the surrounding community yet the miners had higher rates of symptoms compatible with TB, higher rates of risk factors associated with TB disease ( previous lung disease, HIV infection, diabetes mellitus, silicosis and smoking ). Something just does not seem to be right - did the study involve the right number of people (there is no information on how the sample size was determined), could the study have studied well miners because the sick or sicker miners were away from the workplace?

4. There is need for the manuscript to be edited to correct grammatical errors that are spread out throughout the manuscript

5. Line 170 - delete " before eating". It is understood that fasting means before eating .

6. Many statements need to be referenced - including line 185 ( reports of subclinical TB), line 218-219, line 223-224 ( children dropping out of school to seek work in mines), line 225 (proportion of miners with silicosis)

Reviewer #2: (No Response)

7. PLOS authors have the option to publish the peer review history of their article (what does this mean?). If published, this will include your full peer review and any attached files.

Reviewer #1: **Yes: **Jeremiah Chakaya

Reviewer #2: No

---

## [Author Response · Author response to Decision Letter 1]

24 Aug 2022

Reviewer #1: While the authors have responded to all the issues raised in the previous reviews, there are still a number of issues that need to be addresses. These include:

1. Line 59- the authors may wish to use the most recent TB notification data from the 2021 WHO global TB report

Response: Page 3, line 60-61

The current notification as per WHO Global Tuberculosis Report for 2021 has been used. The previous reference i.e., the NTLP Annual Report for 2018 was used as it was the latest readily accessible local (Tanzania) document providing the notification rate of TB, this has been replaced by the WHO Global Tuberculosis Report for 2021

2. Line 92-97- it is advised that the statement be reworded . The study aimed at determining the prevalence of TB and the risk factors for disease in the studied population. It is does not appear appropriate to include the statement that says the study met its aims and will feed into policy. This issue of whether the study met its aims etc can be shifted to the discussion and conclusion section.

Response: Shifted From Page 4, line 100 to Page 23, line 489-492

The phrase has been shifted to the discussion part of the manuscript

3. The findings of the study are significantly different from those of multiple other studies on TB in miners. The issue is not sufficiently discussed. The hypothesis put forward is that there have been a TB screening program among the miners that may have reduced the burden of TB in this population compared to the surrounding community. It is not indicated how long the TB screening program has gone on and what the coverage has been. To begin to see a drop in the burden of TB in a population arising from TB screening and testing services, the screening needs to reach a high proportion of the target population and likely be repeated many times over time. While this issue was not raised in the previous review, it seems important enough to be raised now. The study found a rate of TB in those engaged directly in mining work that was lower than in the surrounding community yet the miners had higher rates of symptoms compatible with TB, higher rates of risk factors associated with TB disease ( previous lung disease, HIV infection, diabetes mellitus, silicosis and smoking ). Something just does not seem to be right - did the study involve the right number of people (there is no information on how the sample size was determined), could the study have studied well miners because the sick or sicker miners were away from the workplace?

Response:

Page 19, line 388-390; Page 20, line 391-405; Page 21, line 422-427

The TB screening programs have been undertaken for about 5 years (mentioned on page 19, line 387). Clarification has been added under discussion part, including that these programs of health education and disease screening are being done by the Kibong’oto Infectious Diseases Hospital (KIDH). The key factor presumed to be the reason for the discrepancy between being presumptive of TB and the burden of TB among the miners was the history of previous lung diseases which included Chronic Obstructive Pulmonary Disease (discussed in another paper), TB and silicosis. As there is overlapping of the symptoms between TB patients and those with other pulmonary disease conditions including COPD and silicosis, this means that, a significant number of miners will have symptoms of TB but most likey they don’t have TB. This has also been clarified under discussion.

Page 6, line 148 - 163

The current report forms just part of the larger study report in which there are other research questions to be answered including primary objective on determination of the prevalence of silicosis and associated risk factors among miners. The sample size of 660 (330 miners and 330 peri-mining communities) is correct. This was calculated based on the observed prevalence of silicosis of 16% as per clinical data from the Occupational Health Clinic Center (OHSC) at KIDH and from a number of outreach disease (including silicosis). screening programs conducted in Mererani mines by KIDH. Both at the OHSC and outreach services, all attended miners are being investigated of silicosis regardless of presence or absence of symptoms. 

4. There is need for the manuscript to be edited to correct grammatical errors that are spread out throughout the manuscript

Response: 

The manuscript has been edited by a native English speaker (the version with track changes cleared)

5. Line 170 - delete " before eating". It is understood that fasting means before eating .

Response: Page 8, line 196

The phrase has been deleted

6. Many statements need to be referenced - including line 185 ( reports of subclinical TB), line 218-219, line 223-224 ( children dropping out of school to seek work in mines), line 225 (proportion of miners with silicosis)

Response:

Reference on the subclinical TB has been added (Page 8, line 211). The phrase about children dropping out of school to seek work in mines has been deleted as this was based on personal communication with mine’s managers and individuals at Mererani mines (Page 10, line 251). About proportion of miners with silicosis, the sentence has been edited with deletion of the phrase showing the significant proportion of miners being observed to have acute silicosis (as this was based on clinical observation at Kibong’oto Infectious Diseases Hospital where the clinical evaluation and disease investigations of the study participants were done, hence the information not readily available from search engines), but retaining the information explaining grouping of the forms of silicosis based on reviewed literature, with references added (Page 10, line 252 – 255)

---

## [Decision Letter · Decision Letter 2]

29 Dec 2022

Prevalence and factors associated with tuberculosis among the mining communities in Mererani, Tanzania

PONE-D-22-00417R2

Dear Dr. Mbuya,

We’re pleased to inform you that your manuscript has been judged scientifically suitable for publication and will be formally accepted for publication once it meets all outstanding technical requirements.

Kind regards,

Billy Morara Tsima, MD MSc

Academic Editor

PLOS ONE

Additional Editor Comments (optional):

Reviewers' comments:

Reviewer's Responses to Questions

**Comments to the Author**

1. If the authors have adequately addressed your comments raised in a previous round of review and you feel that this manuscript is now acceptable for publication, you may indicate that here to bypass the “Comments to the Author” section, enter your conflict of interest statement in the “Confidential to Editor” section, and submit your "Accept" recommendation.

Reviewer #2: (No Response)

Reviewer #3: All comments have been addressed

2. Is the manuscript technically sound, and do the data support the conclusions?

Reviewer #2: (No Response)

Reviewer #3: Yes

3. Has the statistical analysis been performed appropriately and rigorously? 

Reviewer #2: (No Response)

Reviewer #3: Yes

4. Have the authors made all data underlying the findings in their manuscript fully available?

Reviewer #2: (No Response)

Reviewer #3: Yes

5. Is the manuscript presented in an intelligible fashion and written in standard English?

Reviewer #2: (No Response)

Reviewer #3: Yes

6. Review Comments to the Author

Reviewer #2: (No Response)

Reviewer #3: This is a revised version of previously submitted manuscript. A few suggestions to further improve the manuscipt:

1. Title should be "Prevalence...pulmonary tuberculosis..." since the authors just studied pulmonary tuberculosis.

2. Should explicitly state diagnosis criteria in Text, e.g,. in Materials and Methods, stating confirmatory diagnosis was made by GeneXpert MTB/RIF, and briefly discuss the advantages and insufficiency of just using this method as confirmatory diagnosis.

3. There are still typos and grammar errors listed as following. Authors need to check carefully the manuscript.

line 108, "it" should be deleted.

line 120, "but" should be deleted.

line 152, "elaborated" should be elaborate

line 260, "deferred" should be differed

line 266 "5years" should be 5 years

line 392, "but" should be deleted

7. PLOS authors have the option to publish the peer review history of their article (what does this mean?). If published, this will include your full peer review and any attached files.

Reviewer #2: No

Reviewer #3: No

---

## [Editor Report · Acceptance letter]

12 Jan 2023

PONE-D-22-00417R2 

Prevalence and factors associated with tuberculosis among the mining communities in Mererani, Tanzania 

Dear Dr. Mbuya:

I'm pleased to inform you that your manuscript has been deemed suitable for publication in PLOS ONE. Congratulations! Your manuscript is now with our production department. 

Kind regards, 

on behalf of

Dr. Billy Morara Tsima 

Academic Editor

PLOS ONE